# Application of Magnetized Ionized Water and *Bacillus subtilis* Improved Saline Soil Quality and Cotton Productivity

**DOI:** 10.3390/plants13172458

**Published:** 2024-09-02

**Authors:** Zhanbo Jiang, Quanjiu Wang, Songrui Ning, Shudong Lin, Xiaoqin Hu, Zhaoxin Song

**Affiliations:** State Key Laboratory of Eco-Hydraulics in Northwest Arid Region of China, Xi’an University of Technology, Xi’an 710048, China; j_zhanbo@163.com (Z.J.); shudong_lin@163.com (S.L.); 13033758998@163.com (X.H.); songzx9709@163.com (Z.S.)

**Keywords:** soil salinity, soil nutrient, soil microorganism, cotton yield, cotton quality

## Abstract

Soil salinization, a significant global challenge, threatens sustainable development. This study explores the potential of magnetized ionized water irrigation and *Bacillus subtilis* application to mitigate this issue. The former method is hypothesized to enhance soil salt leaching, while the latter is expected to improve soil nutrient availability, thereby increasing microbial diversity. To address the unclear impact of these interventions on soil quality and cotton productivity, this study employs four different experimental methods: magnetized ionized water irrigation (M), application of 45 kg ha^−1^ *B. subtilis* (B), a combination of 45 kg ha^−1^ *B. subtilis* with magnetized ionized water irrigation (MB), and a control treatment with no intervention (CK). This study aims to clarify the effects of these treatments on soil bulk density (BD), field capacity (FC), salinity and alkalinity, nutrient content, microbial activity, and cotton crop yield and quality. Additionally, it aims to evaluate the efficacy of these methods in improving saline soil conditions by developing a soil quality index. The results showed that using magnetized ionized water for irrigation and applying *B. subtilis*, either alone or together, can effectively lower soil pH and salt levels, enhance microbial diversity and abundance, and improve the yield and quality of cotton. Notably, *B. subtilis* application significantly decreased BD and enhanced FC and nutrient content (*p* < 0.05). A correlation was found where soil nutrient content decreased as pH and salt content increased. Furthermore, a strong correlation was observed between the major soil bacteria and fungi with BD, FC, and salt content. Comparatively, M, B, and MB significantly boosted (*p* < 0.01) the soil quality index by 0.21, 0.52, and 0.69 units, respectively, and increased (*p* < 0.05) cotton yield by 5.7%, 14.8%, and 20.1% compared to CK. Therefore, this research offers eco-friendly and efficient methods to enhance cotton production capacity in saline soil.

## 1. Introduction

Soil salinization poses a significant challenge to agricultural development worldwide. Numerous soil-related issues, like decreased nutrient levels and salt stress, result in lower crop production. Saline soils currently cover ~1 billion hectares, or 10% of the earth’s land surface, with a 1% annual increase rate [1]. By 2050, salinization is predicted to affect over half of the world’s arable land [2]. China ranks third in saline soil coverage, with nearly 99 million hectares affected [3].

Cotton is a cornerstone of the global textile industry and ranks among the most important cash crops worldwide. It is the primary source of natural fiber for textile production and plays a crucial role in international trade [4]. Cotton significantly contributes to the GDP of countries such as China, where cotton farming and textile manufacturing are key sectors. In Xinjiang, China, cotton is the principal economic crop, cultivated over 2.4 million hectares, which constitutes 85% of the country’s cotton cultivation and accounts for 91% of national cotton production [5].

However, the Xinjiang region, where saline soils comprise 40% of the national total and saline farmland accounts for 38% of the cultivable area, is particularly affected [6]. The widespread and severe soil salinization in Xinjiang poses significant challenges to cotton farming. The complex effects of soil salinization on soil health and crop productivity often render single management strategies ineffective. Therefore, a multifaceted approach employing various management techniques to address saline soils from multiple angles and mitigate soil salinity is crucial for advancing saline tolerant agriculture.

Activated water technology, particularly magnetized ionized water (MIW), has garnered significant attention for its potential to reduce soil salinity and alkalinity in agricultural irrigation, owing to its environmental sustainability and effectiveness. The process commences with the magnetization and ionization of irrigation water, which weakens the hydrogen bonds among water molecules [7]. This disruption decreases the surface tension and viscosity of the water [8], thereby enhancing its infiltration capacity and the soil salinity and alkalinity leaching efficiency. Mostafazadeh et al. [9] revealed lower mean soil cations and anions at various soil depths when using magnetized irrigation water, compared to non-magnetized water. This indicates that magnetized ionized water can effectively reduce soil salinity. Zhang et al. [10] found that using magnetized water for irrigation led to a 16.6% increase in Na^+^ leaching and a 14.5% increase in Cl^−^ leaching. Moreover, Wei et al. [11] noted a decrease in salt content (12.8–65.0%) in the 40 cm soil layer when using ionized water for irrigation instead of non-ionized water. Water is crucial for dissolving soil minerals and organic materials and is essential for microbial activities. The use of magnetized ionized water provides a foundation for other improvement measures to effectively ameliorate saline soil.

To enhance the nutrient content and efficacy in saline soils, applying plant growth-promoting rhizobacteria (PGPR) has emerged as an efficient and eco-friendly strategy. Soil microorganisms critically influence soil ecosystems, affecting nutrient cycling, soil quality, and plant productivity. They enhance soil fertility by decomposing organic matter, releasing essential nutrients, and participating in nutrient cycling and soil structure formation [12]. Among them, Actinobacteria and Proteobacteria are primary contributors to soil multi-nutrient cycling [13]. However, some microorganisms’ functionality is limited under salt–alkali stress conditions. *B. subtilis*, which secretes stress-resistant spores, maintains its activity during prolonged salt–alkali stress. This bacterium significantly contributes to soil nutrient availability by facilitating nitrogen fixation and phosphorus solubilization. Furthermore, *B. subtilis*’s decomposition of organic nutrients increases the availability of inorganic nutrients in the soil. Yang et al. [14] demonstrated that the use of *B. subtilis* resulted in a notable increase in the alkaline nitrogen content in the soil. This enhancement was positively correlated with the quantity of *B. subtilis* applied (*p* < 0.05). Li et al. [15] further noted that the use of *B. subtilis* decreased soil pH by 7.3% and increased available phosphorus content by 11.2%, compared to untreated soil. Additionally, Wang et al. [16] noted that the use of *B. subtilis* increased the Olsen-P levels in the soil from 14.7 to 23.4 mg kg^−1^ and also made 16.4 mg kg^−1^ of Occluded-P soluble after a 14-day period of incubation. Thus, *B. subtilis* serves as an environmentally friendly soil amendment, enhancing soil nutrient content and diminishing the need for chemical fertilizers.

Various studies have investigated the impacts of treatments such as magnetized ionized water irrigation and *B. subtilis* application on soil characteristics and crop production [11,17,18,19,20]. However, the combined effects of multiple regulatory measures and the interaction mechanisms between soil quality and crop production are increasingly being recognized as crucial for sustainable agricultural development and agroecosystem management [21]. This study concentrates on water and soil, two vital elements of the cotton growth environment. By employing magnetized ionized water and *B. subtilis*, to ameliorate saline soil, we propose that their combined application can enhance soil quality and cotton production. This study’s objectives are as follows: (i) to examine the impact of magnetized ionized water irrigation and *B. subtilis* on soil physical properties, salinity and alkalinity, nutrient content, and microbial (bacteria and fungi) characteristics; (ii) to investigate the correlations among soil physical properties, salinity and alkalinity, nutrients, and microorganisms, and to develop a soil quality evaluation model to assess the improvement’s effectiveness; and (iii) to investigate the impact of the soil quality index on cotton production, taking into account both cotton yield and quality.

## 2. Results

### 2.1. Soil Properties

#### 2.1.1. Soil Physical Properties

Figure 1 illustrates that the application of *B. subtilis* significantly reduced soil bulk density and significantly increased field capacity (*p* < 0.05), while magnetized ionized water had no significant effect (*p* > 0.05). Compared to the treatment without the application of B. subtilis, the average reduction in soil bulk density was 7.0% and the average increase in field capacity was 7.7% when B. subtilis was applied.

#### 2.1.2. Soil Salinity and Alkalinity

Regarding soil salinity and alkalinity, B had a greater impact than M on lowering soil pH and salt content, with MB showing the most significant effect (Figure 2). A significant decrease (*p* < 0.05) in soil pH was observed under MB compared to CK, with a 0.14 reduction. While M and B did not significantly affect pH levels individually, their effects were noticeable. For soil salinity, B and MB treatments resulted in significant reductions (*p* < 0.05) of 7.6% and 16.3%, respectively, compared to CK. Thus, combining *B. subtilis* with magnetized ionized water irrigation is a powerful method for mitigating soil salinity and alkalinity.

#### 2.1.3. Soil Nutrient Content

The treatments’ effectiveness on soil nutrients, excluding soil organic matter content, follows the following order: B > MB > CK > M (Figure 3). However, the impact on ${\mathrm{NH}}_{4}^{+}-N$ and total P content was insignificant. The M treatment demonstrated the lowest nutrient content, yet it was not statistically different from CK, except for available phosphorus. This could be due to the decrease in soil salinity and alkalinity under magnetized ionized water irrigation, potentially enhancing nutrient uptake by cotton plants and reducing residual soil nutrient content. Similar patterns were observed in B and MB treatments, with the B treatment showing a significant improvement in soil nutrient content. Compared to CK, the B treatment significantly increased (*p* < 0.05) soil ${\mathrm{NO}}_{3}^{-}-N$, available P, available K, and organic matter levels by 27.7%, 21.0%, 18.6%, and 28.6%, respectively. This highlights *B. subtilis*’s positive effect on soil nutrient enhancement and availability, as indicated by the increased Available P/Total P ratio.

### 2.2. Diversities of Bacterial Communities

The regulatory measures significantly affected the α diversity of soil bacteria (*p* < 0.05) (Figure 4a). M significantly increased (*p* < 0.05) the Simpson and Shannon indices of soil bacteria by 17.8% and 20.2%, respectively, compared to CK. This suggests that magnetized ionized water irrigation improves bacterial community diversity and species distribution uniformity. *B. subtilis* application significantly boosted soil bacterial growth and α diversity. Compared to CK, the Chao1, Simpson, Shannon indices, and Observed features treated with B increased by 14.8%, 20.2%, 18.6%, and 15.0%, respectively. MB processing showed similar increases of 16.2%, 20.2%, 19.3%, and 16.4%, respectively. These findings suggest that *B. subtilis* application enhances low abundance species in the soil bacterial community, thereby improving community diversity and species distribution uniformity. Using the Bray Curtis similarity coefficient, we conducted a non-metric multidimensional scaling (NMDS) analysis on soil bacterial communities under different regulatory measures (Figure 4b). The NMDS map showed that the four replicates of each regulatory measure were clustered together, indicating good repeatability of the bacterial community structure samples and a low stress coefficient (0.072). At the ASV level, the bacterial communities in B and MB significantly differed from CK, indicating that *B. subtilis* application significantly alters soil bacterial communities.

The taxonomic analysis results showed that 11,972 ASVs were obtained from 16 samples selected from the four regulatory measures. These ASVs were divided into 48 phyla, where the main bacterial communities in the soil under different regulatory measures were similar, yet significant differences in relative abundance were observed. Figure 5 illustrates the composition of the top 10 bacterial phyla with relative abundance in the soil under different regulatory measures. The dominant phyla identified were Proteobacteria (34.1–39.4%), Actinobacteriota (9.4–12.7%), Firmicutes (5.5–15.5%), and Acidobacteriota (6.9–11.3%). The impact of various regulatory measures on the relative abundance of Proteobacteria and Actinobacteriota was insignificant. However, the introduction of *B. subtilis* not only increased the relative abundance of Firmicutes, to which *B. subtilis* belongs, but also potentially influenced the microbial community structure. Specifically, B and MB treatments increased the relative abundance of Firmicutes by 139.4% and 128.3%, respectively, compared to CK. This suggests that the introduction of *B. subtilis* may have cascading effects on the overall microbial composition. The soil bacterial community was further categorized into 822 genera. Figure 6 displays the relative abundance composition of the top 10 soil bacterial genera under different regulatory measures. The dominant genera identified were *Bacillus* (2.0–8.1%), *Pseudomonas* (1.6–4.1%), *Methylophaga* (0.3–7.2%), and *Streptomyces* (1.6–3.2%). The B treatment significantly impacted (*p* < 0.05) the relative abundance of *Bacillus* compared to CK, increasing it by 235.7%, while the M treatment did not. Additionally, the MB treatment significantly affected (*p* < 0.05) the relative abundance of *Bacillus* compared to B, increasing it by 21.9%. These observations indicated that the applied *B. subtilis* successfully colonized the soil, and magnetized ionized water significantly promoted the relative abundance of *Bacillus*.

### 2.3. Diversities of Fungal Communities

The regulatory measures significantly affected the α diversity of soil fungi (*p* < 0.05) (Figure 7a). M significantly increased the Chao1, Simpson, and Shannon indices of soil fungi (*p* < 0.05), showing increases of 36.1%, 30.4%, and 37.1%, respectively, compared to CK. This improvement is attributable to magnetized ionized water irrigation, which achieves superior fungal community diversity and species distribution uniformity. *B. subtilis* application significantly boosted soil fungal α diversity. Compared to CK, the Chao1, Simpson, and Shannon indices and the Observed features treated with B increased by 42.0%, 27.8%, 29.1%, and 36.3%, respectively. MB processing further augmented these increases to 65.1%, 26.6%, 36.2%, and 61.8%, respectively. These findings suggest that *B. subtilis* application enriches low-abundance species in the soil fungal community, thereby enhancing community diversity and species distribution uniformity. The NMDS map (Figure 7b) illustrates that the four replicates of each vegetation reclamation type clustered together, indicating good repeatability of the fungal community structure samples in this study, with a minimal stress coefficient (0.042). At the ASV level, the fungal communities in B and MB distinctly separated from CK, indicating that *B. subtilis* application significantly alters soil fungal communities.

Taxonomic analysis yielded 3573 ASVs from 16 selected samples across four regulatory measures, categorized into 15 phyla. Despite similar main fungal communities in soil under different regulatory measures, significant variations in relative abundance were observed. The top 10 fungal phyla, as depicted in Figure 8, constituted the majority of the soil fungal community. The dominant phyla were Ascomycota, Basidiomycota, Mortierellomycota, and unclassified_fungi, with relative abundances of 53.3–76.2%, 1.7–6.8%, 4.3–6.5%, and 1.8–2.8%, respectively. Regulatory measures did not significantly impact the relative abundance of Ascomycota and unclassified_fungi. However, *B. subtilis* significantly affected the relative abundance of Basidiomycota, reducing it by 68.8% compared to CK and by 74.1% compared to MB. This suggests that *B. subtilis* application may inhibit the relative abundance of Basidiomycota. The soil fungal community was further divided into 202 genera. In Figure 9, the top 10 soil fungal genera’s relative abundance is shown under various regulatory measures. Among them, *Pseudeurotium*, *Botryotrichum*, *Nectria*, and *Hormiactis* were the dominant genera, with relative abundances of 13.0–37.0%, 3.5–13.6%, 2.2–7.9%, and 3.3–6.2%, respectively. Notably, B and MB treatment significantly increased the relative abundance of *Pseudeurotium* compared to CK and M by an average of 235.7%, indicating a significant promotional effect of *B. subtilis* application on the relative abundance of *Pseudoeurotium*.

### 2.4. Relationship between Bacterial and Fungal Communities and Soil Properties

The examination shows intricate connections between soil properties and the composition of microbial communities (Figure 10). Soil bulk density (BD) and field capacity (FC) are negatively correlated. Specifically, soil pH and salt content (SC) have an inverse effect on the nutrient levels in the soil, including ${\mathrm{NH}}_{4}^{+}-N$, ${\mathrm{NO}}_{3}^{-}-N$, available P (AP), total P (TP), available K (AK), and soil organic matter (SOM). The top four bacterial genera in relative abundance (*Bacillus*, *Pseudomonas*, *Methylphaga*, and *Streptomyces*) and the top four fungal genera (*Pseudeurotium*, *Botryotrichum*, *Nectria*, and *Hormiactis*) were selected for the bacterial and fungal matrices in the Mantel test. The test results demonstrate strong correlations (r > 0.5) between dominant bacterial and fungal genera and soil BD, FC, pH, SC, and SOM. Notably, significant correlations (*p* < 0.05) were observed between BD, FC, SC, and SOM with dominant bacterial genera, as well as between BD, FC, and SC and dominant fungal genera (*p* < 0.05). These findings suggest a broader ecological phenomenon where soil structure, salinity and alkalinity, and organic matter influence microbial abundance. The results thus imply potential strategies for enhancing microbial activity, such as managing soil structure and salinity and alkalinity and enriching organic matter.

### 2.5. Soil Quality Index

The dimensionless values of soil indicators, standardized from their determined values, underwent PCA analysis, yielding two principal components with eigenvalues exceeding 1, and accounting for a cumulative variance contribution rate of 78.9%. The distinct distribution of four regulation measures across different quadrants suggested significant individual effects on soil quality (Figure 11). Principal component 1 (PC1), with a variance contribution rate of 64.8%, showed higher loadings for BD, FC, SC, AP/TP, SOM, and bacterial and fungal ASVs. Principal component 2 (PC2), contributing 15.3% variance, revealed higher loadings for pH, ${\mathrm{NH}}_{4}^{+}-N$, ${\mathrm{NO}}_{3}^{-}-N$, AP, AK, and bacterial and fungal Shannon index. Selecting soil indicators with high loadings in both components (absolute values exceeding 90% of the highest factor loading) led to a minimal dataset (MDS) of key soil quality indicators (Figure 12). These key indicators (BD, FC, SC, SOM, ${\mathrm{NO}}_{3}^{-}-N$, AP, bacterial ASVs, and fungal ASVs and Shannon index) formed the basis for calculating the SQI (Figure 13). All regulation measures significantly impacted (*p* < 0.01) the SQI compared to CK, with the M, B, and MB treatments increasing it by 0.21, 0.52, and 0.69 units, respectively.

### 2.6. Cotton Yield Component and Quality

This study uncovered statistically significant effects (*p* < 0.05) of various regulatory measures on cotton yield and quality, emphasizing the crucial role of management strategies in cotton production (Table 1). Specifically, M, B, and MB demonstrated substantial improvements in both yield and quality, surpassing CK. Compared to CK, cotton yield increased significantly under M, B, and MB treatments by 5.7%, 14.8%, and 20.1%, respectively. Similarly, the number of bolls per plant saw significant increases of 8.7%, 6.6%, and 32.7%, respectively, under the same treatments. This suggests a synergistic effect when employing magnetized water and *B. subtilis* together, thereby highlighting the potential of integrated management practices in cotton cultivation. Moreover, cotton quality also exhibited improvements. This was evident from the upper half mean length of cotton fibers increasing by 0.1%, 7.8%, and 17.5% for M, B, and MB treatments, respectively. Furthermore, the uniformity index and specific breaking strength improved across all treatments. In contrast, the micronaire value decreased across all treatments, with the MB treatment showing a notable 6.3% reduction.

A partial least squares path analysis model incorporating variables such as soil physical properties, salinity and alkalinity, nutrients, microorganisms, and cotton yield and quality was developed, as illustrated in Figure 14. The analysis demonstrates that soil physical properties, nutrients, and microorganisms exert a positive influence on both cotton yield and quality, whereas soil salinity and alkalinity negatively affect these parameters. The hierarchy of influence on cotton yield is as follows: soil salinity and alkalinity > soil nutrients > soil microorganisms > soil physical properties. Conversely, the hierarchy of influence on cotton quality is as follows: soil salinity and alkalinity > soil microorganisms > soil nutrients > soil physical properties. A linear regression model was constructed to elucidate the correlation between soil quality and both cotton yield and quality (Table 2). In this model, relative cotton yield and relative quality (Equations (5) and (6)) were designated as dependent variables, whereas the soil quality index (SQI) was defined as the independent variable. Upon establishing the model, the slope was computed, revealing a relationship where yield surpasses quality. This implies that the soil quality index exerts a more significant enhancement effect on cotton yield compared to cotton quality.

## 3. Discussion

### 3.1. Effects of Magnetized Ionized Water on Soil Properties and Crop Production

Magnetization and ionization can alter the physical properties of irrigation water, such as the surface tension and viscosity coefficient. Ding et al. [22] found that magnetization increases water viscosity due to the influence of a magnetic field. This alteration stems from changes in hydrogen bonds within water molecules, affecting their interaction and modifying water’s properties. Liu et al. [23] discovered that magnetized water can experience a surface tension reduction of up to 25%. These property changes improve the infiltration capacity of irrigation water. Al et al. [24] suggested that magnetized water has a collectively beneficial impact on the infiltration properties of soil. Khoshravesh et al. [25] observed that the total water penetration rates were 37.6 and 20 cm after 4 h for water treated with magnetization and untreated water in silty loam soil. The eventual infiltration rates were 0.06 and 0.04 cm min^−1^, respectively. Hamza et al. [26] showed that magnetized water reduced leaching times and necessary water amounts to decrease soil salinity and alkalinity to acceptable levels, indicating that magnetized water enhances leaching efficiency and conserves water. Abdul et al. [27] found that using magnetized water for irrigation can increase the EC value and pH of soil leachate by 58.6% and 2.4%, respectively. This method, which employs magnetized ionized water, mitigates soil salinity stress and facilitates crop nutrient absorption. However, this promotional effect, under identical irrigation and fertilization conditions, may result in the M treatment in comparison to CK (Figure 2 and Figure 3). Magnetized ionized water augments soil moisture content, improves mineral element solubility, and reduces soil salinity and alkalinity. These conditions foster a favorable environment for microbial communities, encouraging the proliferation of diverse microorganisms. Furthermore, applying a magnetic field to irrigation water was found to influence soil microbial structure and abundance. [28] Cui et al. [29] found that magnetized water irrigation significantly enhanced the Chao1 and ACE indices by 21.4% and 23.4%, respectively, compared to non-magnetized water treatment. This increase suggests that magnetized water benefits the richness of soil’s bacterial community, thereby altering its structure. Consistent with previous studies, this study demonstrated that using magnetized ionized water for irrigation led to a notable enhancement in the abundance and variety of soil bacteria and fungi. This improvement fosters an optimal soil environment, promoting cotton growth and development while increasing its productivity (Table 1). Corroborating these findings, Zhou et al. [17,30] and Wei et al. [11] demonstrated that using magnetized and ionized water for irrigation led to an average increase in cotton yield of 22.4% and 21.5%, respectively, in field experiments spanning more than two years. To maximize the effectiveness of the magnetized ionized treatment, this research team has summarized long-term experimental results to propose the technical requirements for magnetized ionized treatment [31]. The effective magnetic field strength at the center position of the agricultural magnetic water device’s pipeline should be between 0.3 and 0.4 T, with the shell’s leakage magnetic field strength at ≤ 2 mT and the effective magnetic length along the pipeline being ≥ 60 cm. The magnetic water device should be installed on the main pipeline or the trunk and branch pipelines of the irrigation system, with the front and rear ends connected to the inlet and outlet of the water conveyance pipe, ensuring that the water is effectively magnetized as it flows through the pipe, without the need for additional energy input. The pipeline flow rate should be ≥ 0.5 m s^−1^ and the effective area used by the magnetic device should be within 1000 m^2^. Future studies should involve continuous multi-year experiments to investigate whether long-term irrigation with magnetized ionized water affects the magnetism of soil particles, as well as the effects of magnetized soil particles on the behavior of conventional water in the soil, facilitating further research discoveries.

### 3.2. Effects of B. subtilis on Characteristics of Soil and Yield of Crops

The metabolism of the microbial community can be altered by the interactions between *B. subtilis* and other microbes, leading to changes in structure and function. This suggests that *B. subtilis* influences the broader microbial ecosystem, contributing to microbial diversity [32]. This study further demonstrates that *B. subtilis* application significantly enhances bacterial and fungal community diversity, with notable differences compared to its absence (Figure 6 and Figure 9). At the bacterial genus level, *B. subtilis* significantly increases the relative abundance of *Bacillus*, *Pseudomonas*, and *Methylophaga*, and, at the fungal genus level, *Pseudoeurotium* (Figure 8 and Figure 11). The increased relative abundance of *Bacillus* indicates successful colonization of the soil by the applied *B. subtilis*. This bacterium can produce plant hormones to stimulate crop growth and participate in nutrient cycling by decomposing soil organic matter [19,33]. Microorganisms are required to break down soil organic nutrients into inorganic forms so that crops can absorb and use them. However, high salt concentrations in saline soils can stress many soil microorganisms, limiting their growth and activity. Soil microorganisms, instrumental in decomposing organic matter and cycling nutrients such as nitrogen, phosphorus, and sulfur, contribute to nutrient availability. However, inhibiting these microorganisms decelerated the natural nutrient cycling process, thereby reducing nutrient conversion rates and availability. *B. subtilis*, a bacterium that generates stress-resistant spores, maintains activity under prolonged salt stress [34]. This bacterium enhances soil inorganic nutrient content by decomposing organic nutrients. In addition, *B. subtilis* secretes acidic substances, including organic acids, which dissolve insoluble soil phosphates like calcium phosphate and iron phosphate [35,36], thereby increasing available phosphorus. The current study corroborates this, showing that *B. subtilis* application boosts the proportion of available phosphorus in total phosphorus (Figure 3). Additionally, *B. subtilis* application reportedly reduced soil salt content by secreting extracellular polymers that bind soil particles, which facilitate large aggregate formation, improve soil structure, and promote salt leaching [37]. Zhou et al. [38] reported that applying *B. subtilis* can reduce soil salinity by 11–31% compared to the blank control. Moreover, *Bacillus* species function as biocontrol agents by generating a variety of antimicrobial compounds that target plant pathogens, thereby reducing the requirement for chemical pesticides, lowering the environmental footprint of agriculture, and mitigating the risk of chemical residues in food [39]. Other bacterial genera, including *Pseudomonas*, *Metrophaga*, and *Streptomyces*, also contribute to crop growth promotion, soil nutrient cycling, and biological control [40,41,42,43,44,45,46,47]. Although the fungal genus *Pseudoeurotium*’s role in soil ecosystems is not explicitly defined, its potential ecological significance in soil environments is evident [48,49]. The *B. subtilis* activity in soil decreases soil salinity and pH, increases nutrient content, and fosters a favorable environment for diverse beneficial microorganisms and increases microbial diversity, thereby improving soil quality and cotton production (Table 1). Zhao et al. [50] reported a 6.1% cotton yield increase with *B. subtilis* application. Furthermore, Zhu et al. [51] demonstrated that *B. subtilis* utilization significantly enhances specific breaking strength by 6.9% and cotton fiber length by 4.3%, indicating a significantly improved cotton quality (*p* < 0.05).

### 3.3. Soil Properties and Soil Quality

Soil salt content and pH, considered “master soil variables”, influence various properties. The phenomenon of salt leaching in this study can be explained as illustrated in Appendix A. The soil water content in the 0–40 cm soil layer at the cotton boll stage was ranked as MB > B > M > CK. This suggested that the treatment M alone was effective in enhancing soil water content compared to CK, although less effective than when combined with *B. subtilis*. Moreover, the increased soil water in the 0–40 cm soil layer indicated that there was indeed sufficient water to affect chemical and physical processes in the soil, including salt leaching, as evidenced by the reduction in soil salinity within this layer (Section 2.1.2). Elevated soil pH and salt levels can cause deficiencies in essential nutrients due to reduced solubility and availability. For instance, phosphorus availability decreases in highly alkaline soils as it can form calcium phosphate precipitates [52]. Moreover, microbial processes crucial for organic matter decomposition, nitrogen fixation, and nutrient cycling can be inhibited or altered in saline soils, potentially impairing soil fertility and plant growth [53]. The results of this research also indicated that the levels of soil pH and salt had adverse effects on the availability of nutrients in the soil (Figure 10). Furthermore, there is a notable relationship (*p* < 0.05, Mantel’s r > 0.5) between the salt levels in the soil and the relative abundance of microbes (bacteria and fungi) (Figure 10). In the comparative analysis, bacteria exhibit a higher correlation with soil salt content than fungi. This observation may be attributable to the enhanced ability of fungi to cope with high osmotic pressures arising from elevated concentrations of organic substrates [54]. Rath et al. [55] revealed that fungi were more resistant to salt exposure than bacteria, and this study also demonstrates that fungi contribute more significantly to microbial communities than bacteria (Figure 14). In addition, soil organic matter also significantly impacts (*p* < 0.05) soil microorganisms, primarily because it serves as a major carbon, nitrogen, and energy source. Abundant organic matter promotes microbial growth and reproduction [56] and significantly affects the composition and structure of soil fungal and bacterial communities [57]. By stimulating beneficial microorganisms, organic matter enhances disease suppression and plant growth promotion, which is vital for sustainable agricultural practices [58].

The soil quality index (SQI), a comprehensive measure of soil productivity and health, plays a crucial role in directing environmentally friendly farming techniques [59,60]. Factors such as soil physical properties (BD and FC), salinity and alkalinity, nutrients (nitrogen, phosphorus, potassium, and SOM), and microbial diversity (bacteria and fungi) variably impact soil quality. These elements influence soil fertility and its ability to sustain plant and microbial life, with microorganisms playing crucial roles in the decomposition of organic matter and cycling of nutrients [61,62]. This study employed principal component analysis to identify key soil quality indicators, including BD, FC, SC, SOM, bacterial ASVs, fungal ASVs, ${\mathrm{NO}}_{3}^{-}-N$, AP, and the fungal Shannon index to calculate the soil quality index. The findings indicated that the utilization of magnetized ionized water for irrigation along with *B. subtilis* had a significant impact (*p* < 0.01) on enhancing soil quality (Figure 13). This enhancement may be attributable to the reduction in soil salinity and alkalinity and the increased diversity of soil microorganisms. Notably, SQI under *B. subtilis* was significantly superior to that under magnetized ionized water irrigation (*p* < 0.01). This difference may be due to the increased soil nutrients observed under *B. subtilis*. This outcome demonstrates the synergistic effect of the two methods, which enhanced soil health and promoted its sustainable use. Corroborating this, Grumezescu et al. [63] underscored the role of magnetic materials in boosting microbial activity. This study revealed that the escalation in the SQI had a more pronounced effect on cotton yield than on cotton quality (Table 2). This discrepancy could be attributed to the critical role of major soil nutrients, particularly nitrogen, phosphorus, and potassium, in cotton growth. The increase in soil microbial diversity facilitated soil nutrient cycling and availability, which reduced soil salinity and alkalinity, thereby promoting nutrient absorption in cotton. These nutrients play direct roles in cotton biosynthesis, such as nitrogen’s contribution to protein synthesis, phosphorus’s role in cell division and organ differentiation, and potassium’s enhancement of water regulation and disease resistance. The augmentation of these nutrients significantly boosts cotton growth and boll number, consequently increasing cotton yield. Conversely, while cotton quality also necessitates adequate nutritional support, it is more profoundly influenced by genetic factors and pest control.

## 4. Materials and Methods

### 4.1. Description of the Experimental Site

This field study was conducted in 2023 at the 7th company of the 8th regiment in Alar City, southern Xinjiang, China (N 40°37′32.26′′, E 80°52′57.78′′). Alar City, known for its warm temperate climate and extreme continental arid desert conditions, experiences significant diurnal temperature variations. In the cotton growing season of April to September 2023, the average daily temperature was 23.3 degrees Celsius, and there was a total rainfall of 13.5 mm (Figure 15). The soil properties within a 0–40 cm profile are detailed in Table 3. Prior to sowing, the average electrical conductivity from the top 40 cm of soil was 3.5 dS m^−1^, indicating the soil was moderately saline [64]. The average pre-sowing soil chemical properties across the 0–40 cm profile were alkali-hydrolyzed nitrogen at 70.3 mg kg^−1^, potassium available at 190.1 mg kg^−1^, phosphorus available at 39.3 mg kg^−1^, organic matter at 11.7 g kg^−1^, and pH at 8.0.

### 4.2. Magnetized Ionized System for Irrigation Water Treatment

The canal water, originating from Tianshan Mountains’ meltwater, underwent filtration and pressurization in the pump room before flowing into the field through buried pipelines. At the field pipeline outlet, a 90 mm polyethylene (PE) pipe connected to a magnetized ionized device was installed. The magnetized ionized system (Xi’an Wangkaiyue Metal Products Co., Ltd., Xi’an, China) comprised a magnetized device, ground electrode, and conductor (Figure 16). Utilizing a permanent rubidium magnet ring with a magnetic field intensity of 3000 Gs, the magnetizer achieved a grounding resistance of 5 Ω, connecting the ground bolt to the ground electrode through a wire. The canal water, not passing through the magnetized ionized system, is termed non-magnetized–ionized water (NMIW), while water passing through is magnetized ionized water (MIW) (Figure 17). Magnetization and ionization treatment altered the water surface tension from 66.8 to 64.0 mN m^−1^ in this study.

### 4.3. Field Management and Experimental Design

On 16 April 2023, cotton cultivar Tahe No.2 (*Gossypium hirsutum* L.) was cultivated and subsequently harvested on 20 September 2023. There were 180,000 plants per hectare, each spaced 10 cm apart. This cultivation employed a drip irrigation system and plastic film mulch, as depicted in Figure 18 and Figure 19. The drip irrigation system followed a sequential row spacing pattern of 10 cm (outside narrow), 60 cm (wide), 10 cm (inside narrow), 60 cm (wide), and again 10 cm (outside narrow). Two irrigation lines were placed every six rows, beneath a 1.5 m wide plastic covering with a 50 cm space between each covering. The 1.6 cm diameter drip lines had emitters spaced 30 cm apart. The irrigation scheduling of this experiment was regulated by the local government to ensure the reasonable allocation of water resources in Xinjiang. The irrigation interval ranges from 7 to 10 days. There is no specified amount of water to be used during irrigation, as each irrigation event is restricted by local government regulations to last no longer than 5 h. In this experiment, each irrigation lasted a full 5 h. Irrigation commenced on 6 June 2023, with a total water application of 539 mm recorded. The irrigation scheduling of this experiment is shown in Figure 20.

The experiment comprised four treatments: untreated irrigation (CK), irrigation with magnetized ionized water (M), untreated water irrigation with 45 kg ha^−1^ *B. subtilis* added (B), and irrigation with magnetized water with 45 kg ha^−1^ *B. subtilis* added (MB). The application rate of *B. subtilis* is determined based on the research of Jiang et al. [65]. Every experimental plot was 2 m wide and 30 m long, with each treatment repeated three times in a randomized block layout. *B. subtilis* (LV-01), provided by Shandong Lvlong Biotechnology Co., Ltd. (Weifang City, Shandong Province, China), was in wettable powder form, containing 20 billion viable spores per gram. The initial cotton irrigation event (6 June 2023) utilized pressure fertilizer barrels for its application through the drip irrigation system, as shown in Figure 17.

### 4.4. Measurements and Calculations

#### 4.4.1. The Bulk Density and Field Capacity of Soil

In drip irrigation settings, cotton roots primarily exist in the 0−40 cm soil layer [66], thus this layer was designated as the cotton’s principal root zone for further analysis. Bulk density and field capacity measurements were conducted in both wide and narrow rows (Figure 18) using the cutting ring method [67]. Soil samples were collected at depths of 0, 10, 20, 30, and 40 cm below the surface during the critical boll stage of cotton growth.

#### 4.4.2. The pH and Salinity of the Soil

The depths of soil sampling were 0, 10, 20, 30, and 40 cm. Samples were taken from specific field locations (Figure 18) during the critical boll stage of cotton growth. The cotton boll stage is crucial for determining the yield and quality of cotton. Soil conditions such as soil salinity and alkalinity, nutrient availability, and microbial activity significantly influence the success of cotton production during this stage. Effective management practices tailored to optimize these factors can enhance cotton yield and improve fiber quality. Each extraction hole was carefully refilled to prevent potential experimental discrepancies. Subsequently, the soil samples were dried in a fan-assisted oven at 105 °C for 24 h. Soil pH was analyzed using a PHS-25 pH meter (Shanghai Yidian Scientific Instrument Co., Ltd., Shanghai, China) on a 1:2.5 soil–water extract at 25 °C. Additionally, the electrical conductivity of a soil–water extract with a ratio of 1:5 was determined using a DDSJ-308A conductivity meter under identical temperature conditions. This conductivity value (EC_1:5_, dS m^−1^) was utilized to predict the amount of salt in the soil (SC, g kg^−1^) using a linear relationship (SC = 0.99 × EC_1:5_), supported by a coefficient of determination (R^2^ = 0.999, *n* = 30).

The amount of salt within the 0–40 cm soil layer (*S*, g m^−2^) was calculated in the following formula:(1)$$S=10\bar{\rho }\left(\frac{1}{3}{\mathrm{SC}}_{\mathrm{inside}\mathrm{narrow}}+\frac{4}{21}{\mathrm{SC}}_{\mathrm{wide}}+\frac{1}{3}{\mathrm{SC}}_{\mathrm{outside}\mathrm{narrow}}+\frac{1}{7}{\mathrm{SC}}_{\mathrm{bare}}\right)$$
where *S* represents the soil salt content at the cotton boll stage. The *SC*_inside narrow_, *SC*_wide_, *SC*_outside narrow_, and *SC*_bare_ variables represent the salt content of the soil for the inside narrow, wide, outside narrow, and bare soil strips (g m^−2^), respectively. Proportional weights of 1/3, 4/21, 1/3, and 1/7 were allocated to reflect the relative widths of these strips at their respective locations.

#### 4.4.3. Soil Nutrient

At 10 cm intervals from the surface to a 40 cm depth, soil samples were systematically collected from the center of inside and outside narrow rows at the cotton boll stage. Soil nutrient measurement index methods refer to Bao [68]. Fresh soil samples were tested for nitrogen levels (${\mathrm{NH}}_{4}^{+}-N$ and ${\mathrm{NO}}_{3}^{-}-N$) with a fully automated discrete analyzer (Smartchem450, AMS Alliance, Paris, France). Dried soil samples were used to determine available phosphorus content (AP) through sodium bicarbonate (NaHCO_3_) extraction and the molybdenum antimony anti-colorimetric method. The total phosphorus content (TP) was ascertained using the sulfuric and perchloric acid digestion method. Soil available potassium content (AK) was extracted using 1 mol L^−1^ NH_4_OAc and quantified via flame photometry. Lastly, soil organic matter content (SOM) was determined in dried soil samples using the external heating method with potassium dichromate and concentrated sulfuric acid.

#### 4.4.4. Soil Microbial Properties

At 10 cm intervals from the surface to a 40 cm depth, soil samples were systematically collected from the center of inside and outside narrow rows at the cotton boll stage and combined. Four replicates of each treatment were stored in a refrigerator at −80 °C. Specific primers with barcodes were used to amplify the distinct regions of the 16S rRNA and ITS genes (16SV4, GTGCCAGCMGCCGCGGTAA, GGACTACHVGGGTWTCTAAT; ITS1-5F, GGAAGTAAAAGTCGTAACAAGG, GCTGCGTTCTTCATCGATGC). Each PCR reaction comprised 15 µL of Phusion^®^ High—Fidelity PCR Master Mix (New England Biolabs, Beijing, China), 0.2 µM of forward and reverse primers, and ~10 ng template DNA. The thermal cycling conditions included an initial denaturation at 98 °C for 1 min, followed by 30 cycles of denaturation at 98 °C for 10 s, annealing at 50 °C for 30 s, elongation at 72 °C for 30 s, and a final extension at 72 °C for 5 min. An equal volume of 1× loading buffer (containing Sybgreen) was mixed with the PCR products, and electrophoresis was performed on a 2% agarose gel for DNA detection. The PCR products were mixed in equal proportions, and then a Qiagen Gel Extraction Kit (Qiagen, Hilden, Germany) was used to purify the mixed PCR products. Following the manufacturer’s recommendations, sequencing libraries were generated with NEBNext Ultra II DNA Library Prep Kit (New England Biolabs, Beijing, China). The library quality was evaluated on the Qubit 2.0 fluorometer (Thermo Scientific, Shanghai, China) and the Agilent Bioanalyzer 2100 system (Agilent, Beijing, China). Finally, the library was sequenced on an Illumina NovaSeq (Shanghai, China) platform and 250 bp paired-end reads were generated.

#### 4.4.5. Soil Quality

The soil quality index was computed for the evaluation of soil quality. First, a standardized principal component analysis (PCA) was performed on soil indicators, with a focus on extracting principal components having eigenvalues > 1, each accounting for a minimum of 5% of the variance. The cumulative contribution rate of these principal components exceeded 75% of the variance. Second, within each PC (principal component), key soil quality indices were identified, characterized by absolute factor load values surpassing 90% of the highest factor load. These key indices were retained in the minimum dataset (MDS) and subsequently converted to dimensionless values ranging from 0 to 1. For indicators where higher values are preferable (e.g., ${\mathrm{NH}}_{4}^{+}-N$, ${\mathrm{NO}}_{3}^{-}-N$, total P, available P, available P/total P, available K, and soil organic matter), dimensionless processing was applied using Equation (2). Conversely, for indices favoring lower values (soil salt content and pH), Equation (3) was employed. The final soil quality index was determined using Equation (4) [21].
(2)$$T_{i}=\frac{X_{i}-X_{\min}}{X_{\max}-X_{\min}}$$
(3)$$T_{i}=\frac{X_{\max}-X_{i}}{X_{\max}-X_{\min}}$$
where *T_i_*, *X_i_*, *X*_min_, and *X*_max_ represent the dimensionless values, actual measured values, minimum measured values, and maximum measured values of key soil quality indicators, respectively.
(4)$$\mathrm{SQI}=\overset{n}{\underset{i=1}{\sum }}W_{i}\cdot T_{i}$$
where *W_i_* represents the weight of key soil quality indicators, and *n* indicates the quantity of indicators in MDS.

#### 4.4.6. Cotton Quality

During harvest season, 50 cotton bolls were randomly collected from each experimental plot for analysis. The specific breaking strength, micronaire value, upper half mean length, and uniformity index were evaluated by the Xinjiang academy of agricultural sciences (Urumqi, Xinjiang, China). A cotton fiber testing instrument (1000 M 700, Uster Technologies, Greenville, SC, USA) was utilized to assess fiber quality.

#### 4.4.7. Calculation of Relative Yield and Relative Quality

This study develops a linear model to assess the influence of the SQI on cotton yield and quality. Relative processing for indicators where higher values are desirable (*Z_i_*), such as seed cotton yield, boll number per plant, specific breaking strength, uniformity index, and upper half mean length, was conducted using Equation (5). Conversely, for indicators where lower values are preferable, such as the micronaire value, Equation (6) was utilized.
(5)$$y_{i}=\frac{Z_{i}}{Z_{i,\max}}$$
(6)$$y_{i}=\frac{Z_{i,\min}}{Z_{i}}$$
where *Z_i_* is the cotton yield and quality indicator, and *Z_i_*_, max_ and *Z_i_*_, min_ are the maximum and minimum value of the above indicators; *y_i_* is the relative value of the above indicators.

#### 4.4.8. Analysis of Data

ANOVA analysis was conducted with SAS software (version 9.2). Significant differences among treatments were identified using least significant difference (LSD) test at a significance level of *p* < 0.05. Data analysis and graphical representation were executed using Microsoft Excel 2020 and Origin 2019b, respectively.

## 5. Conclusions

This study focused on the effects of magnetized ionized water and *B. subtilis* on soil physical properties, salinity and alkalinity, nutrient content, and microbial diversity, as well as on a soil quality assessment. The research discovered that the levels of nutrients in the soil, including ${\mathrm{NH}}_{4}^{+}-N$, ${\mathrm{NO}}_{3}^{-}-N$, AP, TP, AK, and SOM, negatively correlate with soil salinity and alkalinity, while the relative abundance of dominant soil bacterial and fungal genera significantly correlates with soil salt content. Compared to CK, the M, B, and MB treatments significantly improved (*p* < 0.01) the SQI by 0.21, 0.52, and 0.69 units, respectively, with a greater promotion effect on cotton yield than cotton quality. This research offers valuable insights into the effectiveness of combining magnetized ionized water irrigation with *B. subtilis* application in saline agriculture. Future studies should prioritize long-term experiments to validate these findings, highlighting the significant benefits of adopting eco-friendly and effective strategies for improving soil quality and cotton yield, thereby promoting sustainable agricultural practices.

## Figures and Tables

**Figure 1 plants-13-02458-f001:**
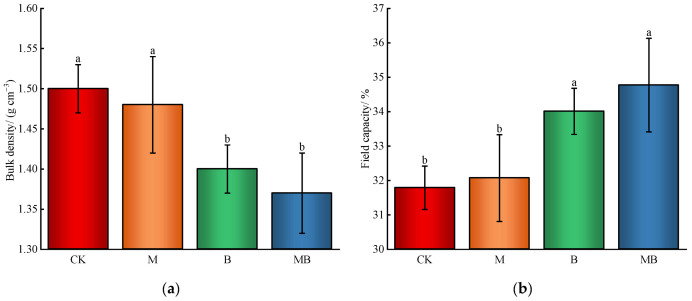
Soil physical properties. Bulk density (**a**). Field capacity (**b**). CK denotes untreated irrigation, M denotes magnetized ionized water irrigation, B denotes untreated water irrigation with 45 kg ha^−1^ *B. subtilis*, and MB denotes magnetized water irrigation with 45 kg ha^−1^ *B. subtilis.* The data represent the average of the three replicates. Different letters above the bars indicate significant differences among treatments at *p* < 0.05.

**Figure 2 plants-13-02458-f002:**
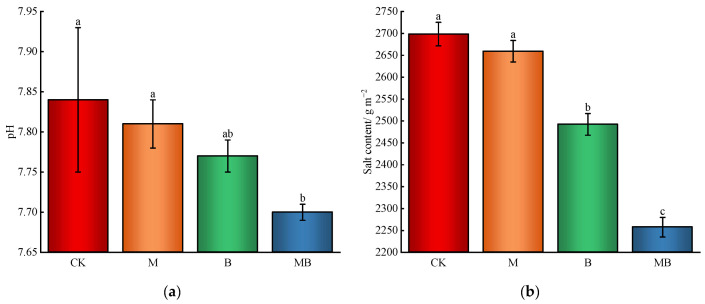
Soil salinity and alkalinity. Soil pH (**a**). Salt content (**b**). CK denotes untreated irrigation, M denotes magnetized ionized water irrigation, B denotes untreated water irrigation with 45 kg ha^−1^ *B. subtilis*, and MB denotes magnetized water irrigation with 45 kg ha^−1^ *B. subtilis.* The data represent the average of the three replicates. Different letters above the bars indicate significant differences among treatments at *p* < 0.05.

**Figure 3 plants-13-02458-f003:**
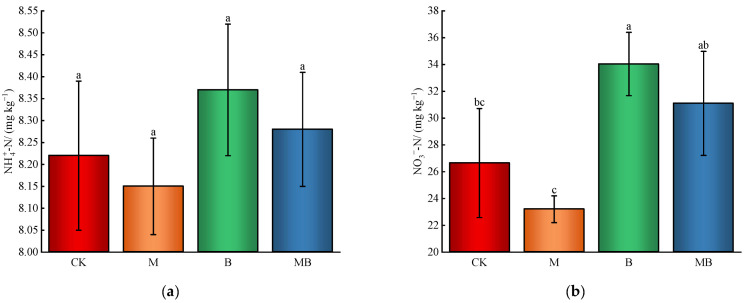
Soil nutrient content. ${\mathrm{NH}}_{4}^{+}-N$ content (**a**). ${\mathrm{NO}}_{3}^{-}-N$ content (**b**). Available P content (**c**). Total P content (**d**). Available P content/Total P content (**e**). Available K content (**f**). Soil organic matter content (**g**). CK denotes untreated irrigation, M denotes magnetized ionized water irrigation, B denotes untreated water irrigation with 45 kg ha^−1^ *B. subtilis*, and MB denotes magnetized water irrigation with 45 kg ha^−1^ *B. subtilis.* The data represent the average of the three replicates. Different letters above the bars indicate significant differences among treatments at *p* < 0.05.

**Figure 4 plants-13-02458-f004:**
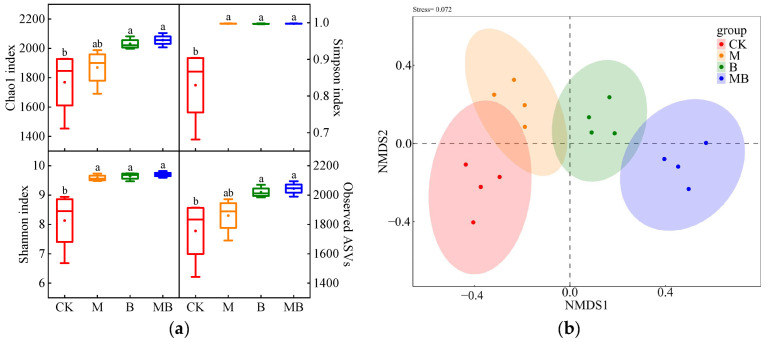
Soil bacterial communities’ *α* diversity (**a**) and NMDS analysis (**b**) under different regulatory measures. CK denotes untreated irrigation, M denotes magnetized ionized water irrigation, B denotes untreated water irrigation with 45 kg ha^−1^ *B. subtilis*, and MB denotes magnetized water irrigation with 45 kg ha^−1^ *B. subtilis.* The data represent the average of the four replicates. Errors bars indicate standard errors. Different letters above the bars indicate significant differences among treatments at *p* < 0.05.

**Figure 5 plants-13-02458-f005:**
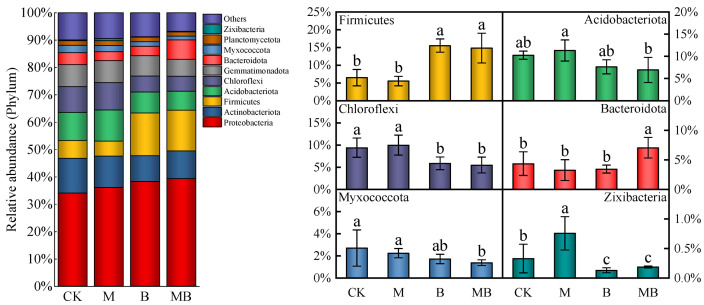
Relative abundance of bacterial community composition at the top 10 phylum levels under different regulatory measures. CK denotes untreated irrigation, M denotes magnetized ionized water irrigation, B denotes untreated water irrigation with 45 kg ha^−1^
*B. subtilis*, and MB denotes magnetized water irrigation with 45 kg ha^−1^ *B. subtilis*. The data represent the average of the four replicates. Errors bars mean standard errors. Different letters above the bars indicate significant differences among treatments at *p* < 0.05.

**Figure 6 plants-13-02458-f006:**
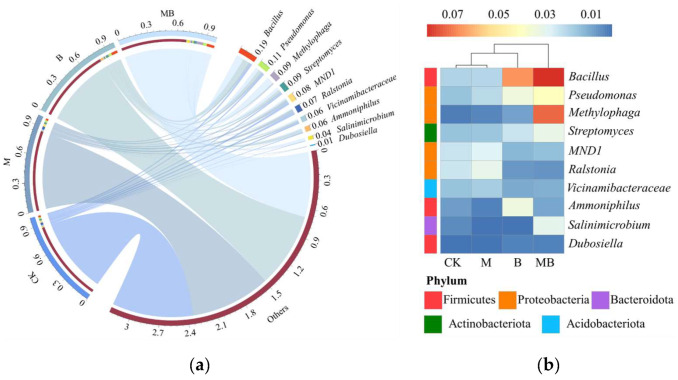
Relationships between dominant bacterial genus and regulatory measures (**a**) and heat map of dominant genus of bacterial communities under different regulatory measures (**b**). CK denotes untreated irrigation, M denotes magnetized ionized water irrigation, B denotes untreated water irrigation with 45 kg ha^−1^ *B. subtilis*, and MB denotes magnetized water irrigation with 45 kg ha^−1^ *B. subtilis.*

**Figure 7 plants-13-02458-f007:**
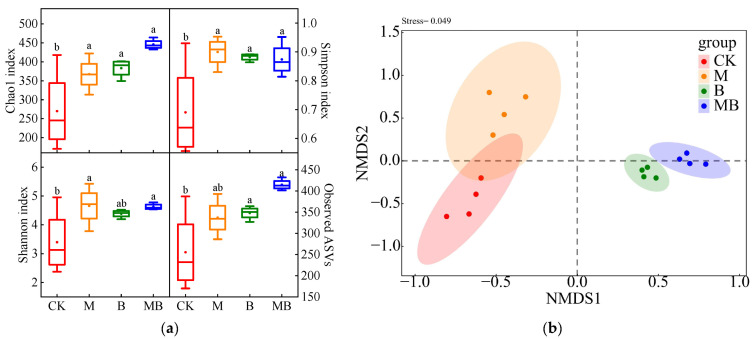
Soil fungal communities’ *α* diversity (**a**) and NMDS analysis (**b**) under different regulatory measures. CK denotes untreated irrigation, M denotes magnetized ionized water irrigation, B denotes untreated water irrigation with 45 kg ha^−1^ *B. subtilis*, and MB denotes magnetized water irrigation with 45 kg ha^−1^ *B. subtilis.* The data represent the average of the four replicates. Errors bars indicate standard errors. Different letters above the bars indicate significant differences among treatments at *p* < 0.05.

**Figure 8 plants-13-02458-f008:**
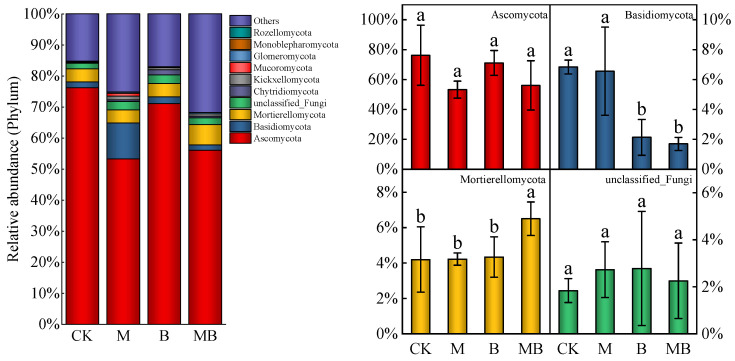
Relative abundance of fungal community composition at the top 10 phylum level under different regulatory measures. CK denotes untreated irrigation, M denotes magnetized ionized water irrigation, B denotes untreated water irrigation with 45 kg ha^−1^ *B. subtilis*, and MB denotes magnetized water irrigation with 45 kg ha^−1^ *B. subtilis.* The data represent the average of the four replicates. Errors bars indicate standard errors. Different letters above the bars indicate significant differences among treatments at *p* < 0.05. “Others” refers to fungal species that have been classified but do not belong to the main groups of interest in our study. “Unclassified fungi” includes fungal sequences that could not be accurately classified at a finer taxonomic level due to limitations in the available database or sequence similarity. These fungi could not be assigned to a specific taxonomic group beyond a broad level.

**Figure 9 plants-13-02458-f009:**
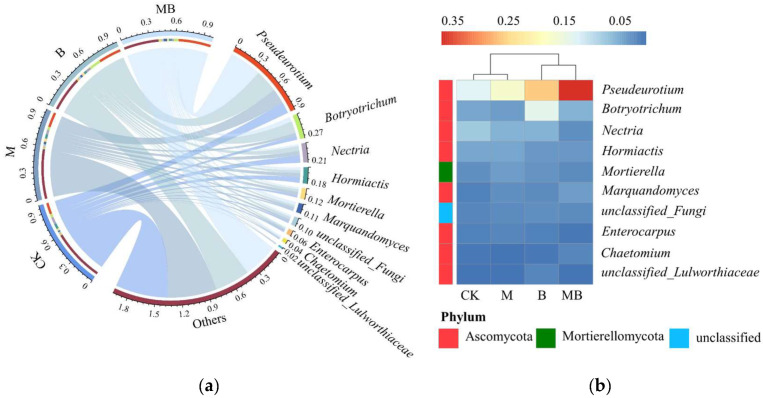
Relationships between dominant fungal genus and regulatory measures (**a**) and heat map of dominant genus of fungal communities under different regulatory measures (**b**). CK denotes untreated irrigation, M denotes magnetized ionized water irrigation, B denotes untreated water irrigation with 45 kg ha^−1^ *B. subtilis*, and MB denotes magnetized water irrigation with 45 kg ha^−1^ *B. subtilis.*

**Figure 10 plants-13-02458-f010:**
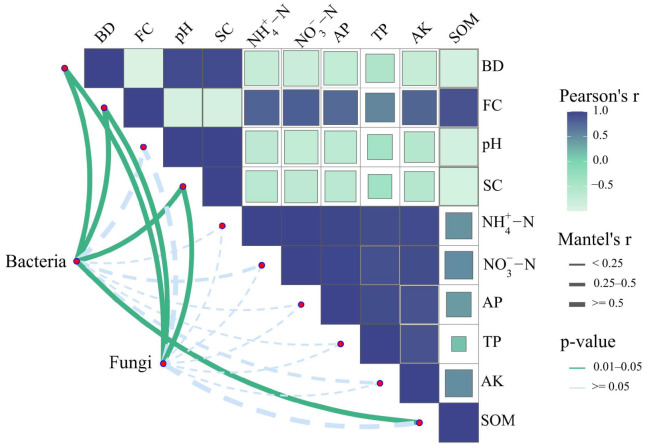
Mantel test analysis of dominant bacterial and fungal genus and soil properties.

**Figure 11 plants-13-02458-f011:**
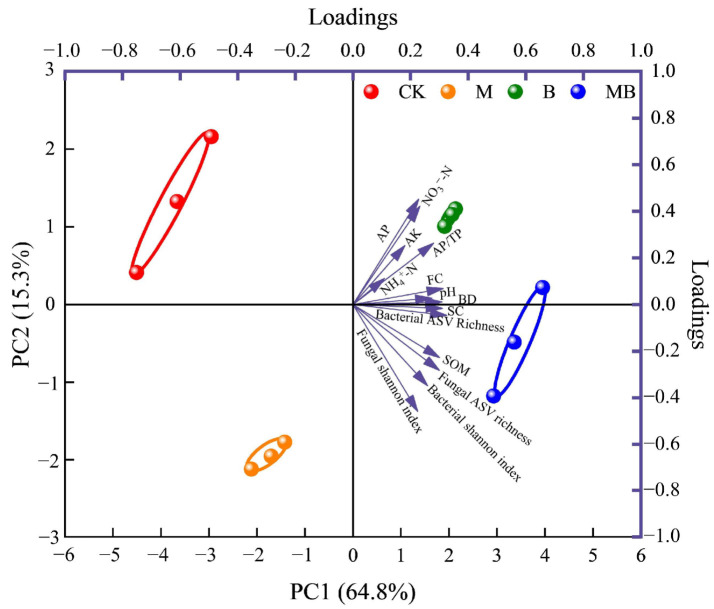
Principal component analysis of soil properties. CK denotes untreated irrigation, M denotes magnetized ionized water irrigation, B denotes untreated water irrigation with 45 kg ha^−1^ *B. subtilis*, and MB denotes magnetized water irrigation with 45 kg ha^−1^ *B. subtilis.*

**Figure 12 plants-13-02458-f012:**
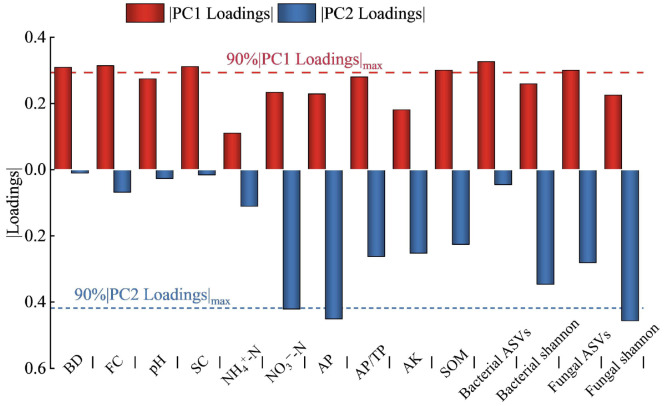
Loading vectors of soil properties.

**Figure 13 plants-13-02458-f013:**
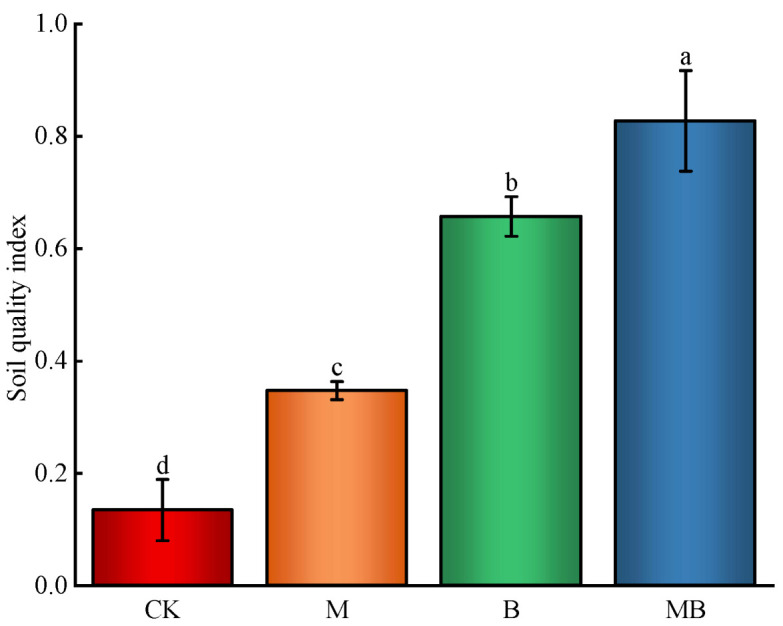
Effects of regulatory measures on soil quality index. CK denotes untreated irrigation, M denotes magnetized ionized water irrigation, B denotes untreated water irrigation with 45 kg ha^−1^ *B. subtilis*, and MB denotes magnetized water irrigation with 45 kg ha^−1^ *B. subtilis.* The data represent the average of the three replicates. Errors bars indicate standard errors. Different letters above the bars indicate significant differences among treatments at *p* < 0.01.

**Figure 14 plants-13-02458-f014:**
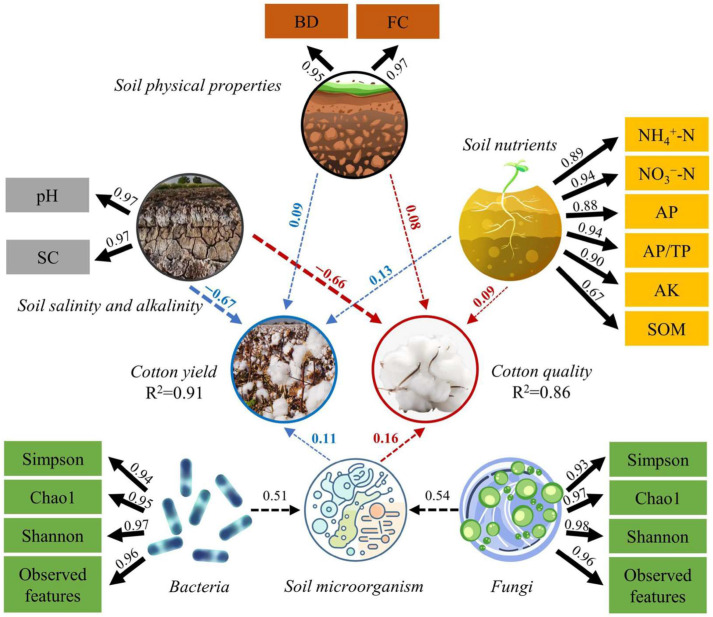
Partial least squares path modeling for the interaction between soil salinity and alkalinity, nutrients, microorganism, cotton yield, and quality.

**Figure 15 plants-13-02458-f015:**
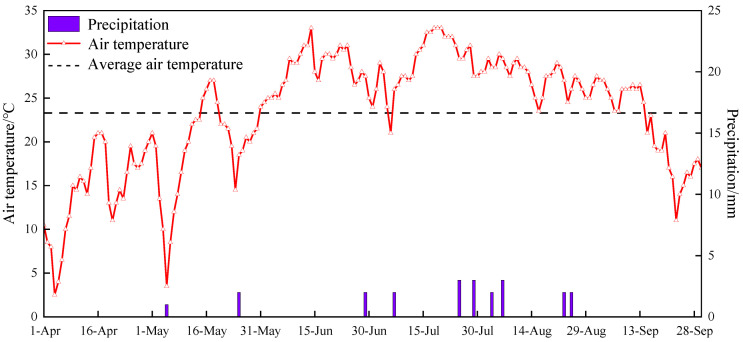
Average daily temperature and rainfall throughout the cotton growth season.

**Figure 16 plants-13-02458-f016:**
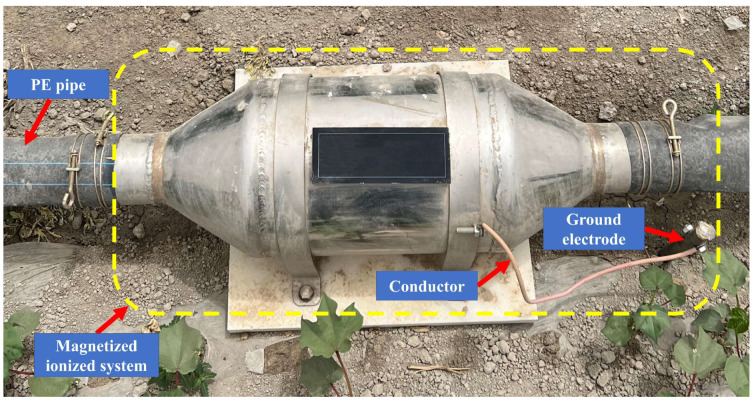
Magnetized ionized system.

**Figure 17 plants-13-02458-f017:**
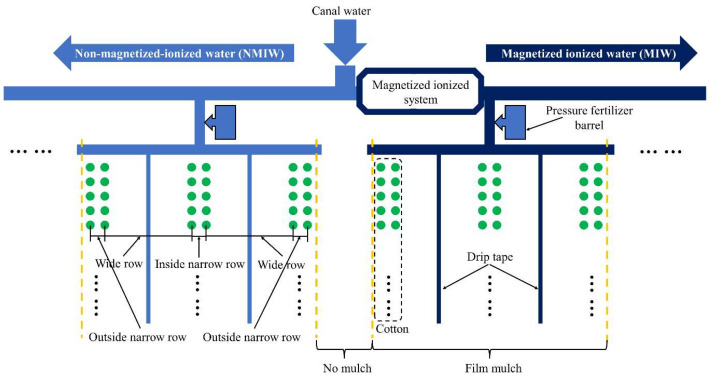
Schematic diagram of the drip irrigation system arrangement of the experiments.

**Figure 18 plants-13-02458-f018:**
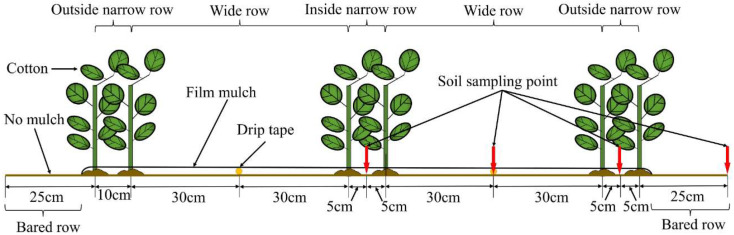
Locations of cotton planting.

**Figure 19 plants-13-02458-f019:**
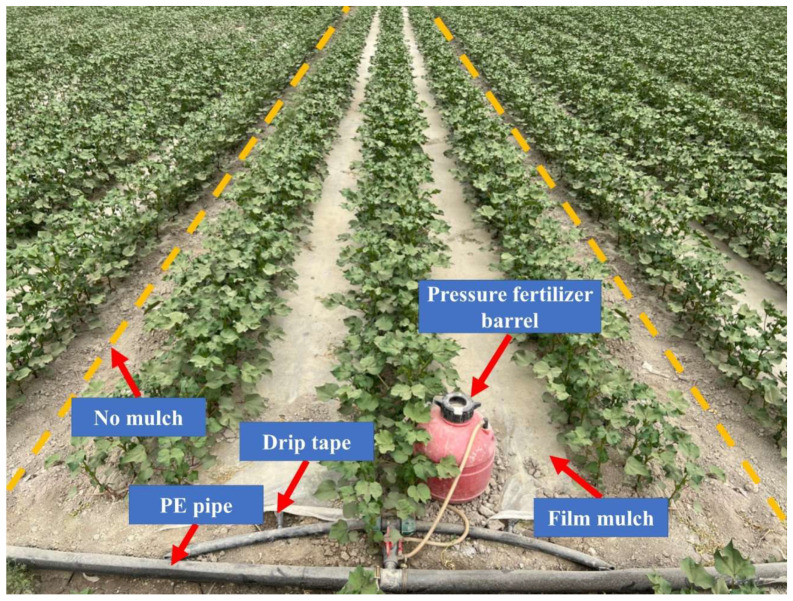
Experimental site.

**Figure 20 plants-13-02458-f020:**
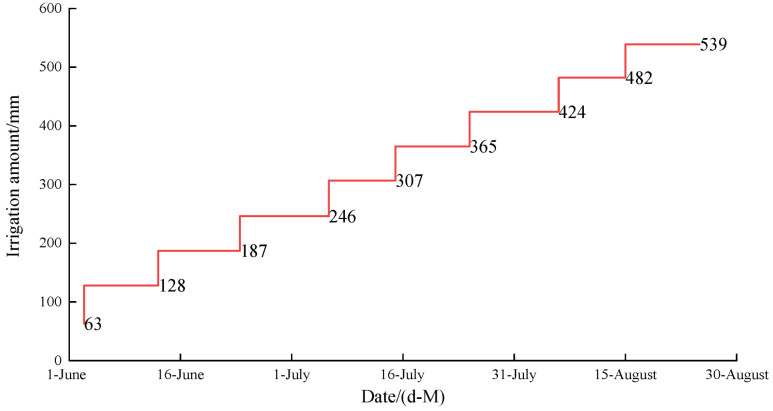
Cotton irrigation scheduling.

**Table 1 plants-13-02458-t001:** Effects of regulatory measures on the production and quality of cotton.

Index	CK	M	B	MB
Seed cotton yield (kg ha^−1^)	6045.77 c ± 36.77	6390.49 bc ± 503.15	6941.02 ab ± 235.97	7260.63 a ± 516.86
Boll number per plant	3.91 b ± 0.20	4.25 b ± 0.25	4.17 b ± 0.11	5.19 a ± 0.41
Upper half mean length (mm)	27.16 c ± 0.53	27.19 c ± 0.39	29.27 b ± 0.51	31.92 a ± 0.43
Uniformity index (%)	77.80 d ± 0.59	80.44 c ± 0.66	82.12 b ± 0.84	85.69 a ± 0.69
Specific breaking strength (cN tex^−1^)	29.09 b ± 1.20	30.73 b ± 0.72	30.41 ab ± 1.84	32.54 a ± 1.02
Micronaire value	5.08 a ± 0.08	4.92 ab ± 0.17	4.90 ab ± 0.16	4.76 b ± 0.10

Note: CK denotes untreated irrigation, M denotes magnetized ionized water irrigation, B denotes untreated water irrigation with 45 kg ha^−1^ *B. subtilis*, and MB denotes magnetized water irrigation with 45 kg ha^−1^ *B. subtilis.* The data represent the average of the three replicates. Different letters within a column indicate significant differences among all treatments at *p* < 0.05.

**Table 2 plants-13-02458-t002:** Relationship between yield and quality of cotton and soil quality index.

Cotton Production Index		Linear Function Model	Average Slope	R^2^	RMSE
Yield	Seed cotton yield	*y_Y_* = 0.24*SQI* + 0.80	0.26	0.99	0.01
	Boll number per plant	*y_B_* = 0.28*SQI* + 0.70		0.66	0.08
Quality	Upper half mean length	*y_L_* = 0.21*SQI* + 0.86	0.13	0.83	0.03
	Uniformity index	*y_U_* = 0.12*SQI* + 0.89		0.94	0.01
	Specific breaking strength	*y_S_* = 0.12*SQI* + 0.88		0.74	0.03
	Micronaire value	*y_M_* = 0.08*SQI* + 0.93		0.88	0.01

**Table 3 plants-13-02458-t003:** Physical characteristics of the soil at the test location.

Soil Layer (cm)	Particle Composition			Soil texture	Bulk Density (g cm^−3^)	Soil Water Content (cm^3^ cm^−3^)		
	Clay (%)	Silt (%)	Sand (%)			*θ* _WP_	*θ* _FC_	*θ* _S_
0–20	12.18	79.11	8.60	Silty loam	1.46	0.042	0.31	0.41
20–40	12.66	81.93	5.42	Silty loam	1.52	0.042	0.33	0.41

*θ*_WP_ is the permanent wilting point; *θ*_FC_ is the field capacity; and *θ*_s_ is the saturated soil water content.

## Data Availability

The data presented in this study are available on request from the corresponding author.

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
