# Peer review of "Application of Magnetized Ionized Water and Bacillus subtilis Improved Saline Soil Quality and Cotton Productivity"

_plants, 2024, doi:10.3390/plants13172458_

Round 1

Reviewer 1 Report

Comments and Suggestions for Authors

The authors tested a combined approach of magnetic ionized water and a bioproduct to increase soil quality, cotton lint productivity, and quality. There are a few points that the authors must fix/clarify to improve the quality of the manuscript. I hope my comments will be helpful:

**Title: OK

**Abstract: OK

**Keywords: OK

1. Introduction

-The current flow is:

Problems caused by soil salinization and increasing in the affected area, the importance of cotton in Xinjiang, and the impact of salinization on cotton production and the need to cope with salinization > Importance of magnetized ionized water (MIW) to leach ions and reduce soil salinity > Use of rhizobacteria to improve plant nutrition and soil microbial communities, especially Bacillus subtilis > Need of combined studies with different methods to increase plant productivity and objectives of this work

The flow is proper, but the first paragraph mixes too many ideas. I suggest the following flow:

Problems caused by soil salinization and increasing in the affected area > Importance of cotton in the World/China/Xinjiang > Impact of salinization on cotton plants and the need to cope with it > Importance of magnetized ionized water (MIW) to leach ions and reduce soil salinity > Use of rhizobacteria to improve plant nutrition and soil microbial communities, especially Bacillus subtilis > Need of combined studies with different methods to increase plant productivity and objectives of this work

2. Results

2.1. Soil properties

2.1.1. Soil physical properties: OK

2.1.2. Soil salinity and alkalinity: OK

2.1.3. Soil nutrient:

-In Table 1, “Different letters within a column indicate significant differences among all treatments at P < 0.05.” In the text, the authors comment on results with P < 0.01. I recommend adding asterisks to the variables in the Index column: * for P < 0.05, ** for P < 0.01, *** for P < 0.001 (if there is any case like this), and ns for non-significant.

2.2. Diversities of bacterial communities

-Lines 135, 139: Typo: “indices” not “index.”

-Lines 179-181: Genus Bacillus is part of the Firmicutes phylum. Please reconsider the relevance of informing that adding bacteria from the Bacillus genus/Firmicutes phylum, increased the number of bacteria from the Bacillus genus/Firmicutes phylum in the samples.

-Figure 2: The right side of the figure is an interesting detail of the left side, but the authors can explore it better. Instead of presenting the variations for only the top-4 phyla, including the phylum that has an expected variation, I suggest presenting data for six phyla with higher variation among the treatments, excluding the Firmicutes phylum because its variation is visible in the left part of the figure.

2.3. Diversities of fungal communities

-Please explain the existence of the groups “Others” and “Unclassified fungi.” What are the differences between them?

2.4. Relationship between bacterial and fungal communities and soil properties: OK

2.5. Soil quality index: OK

2.6. Cotton yield component and quality

-Table 2: Typo: “cN tex-1” not “Cn tex-1.”

-Did the authors observe differences in the lint turnout? Please provide this detail, even if there was no significant difference.

3. 3. Discussion

3.1. Effects of magnetized ionized water on soil properties and crop production: OK

3.2. Effects of B. subtilis on characteristics of soil and yield of crops: OK

3.3. Soil properties and soil quality: OK

4. Materials and methods

4.1. Description of the experimental site: OK

4.2. Field management and experimental design:

-The authors provided enough details about the soil, the plants, and the bacteria but barely mentioned anything about the MIW system. Please provide information (maker, model, settings) as text in this subsection. Writing this information only in Figure 16 may be lost if somebody prints or photocopies the document.

-It is not clear how the authors used the MIW generator. Did the authors use one generator for each drip tape, each block, or another design? Please explain it.

-Please provide the strain of the Bacillus subtilis used in the experiment.

4.3. Measurements and calculations

4.3.1. The bulk density and field capacity of soil: OK

4.3.2. The pH and salinity of the soil: OK

4.3.3. Soil nutrient

-Line 568: The authors wrote “…soil samples were tested for nitrate nitrogen levels…” but presented ammonium (NH4 1+) in the same line. Please explain better what they meant.

-Please provide the maker and model of the fully automated system to measure N.

-Please provide references for the other methods.

4.3.4. Soil microbial properties

-Please provide the maker and model of the used instruments.

-The authors

4.3.5. Soil quality: OK

4.3.6. Cotton quality

-Please provide the maker of the instrument.

-The authors used experimental samples instead of bale samples. Please check the elongation and SFI, even if they are not calibrated measurements. This information may be significant for further research. 

4.3.7. Calculation of relative yield and relative quality: OK

4.3.8. Analysis of Data: OK

5. Conclusion

-The conclusion is fair, but the authors should propose further research work. Another point that I suggest discussing is the availability of using MIW. Even if it has a beneficial effect, how much power does it use? How many generators should be per hectare? The authors cannot discuss this with the data of this work, but they must comment on this weak point to guide further projects/research.

**Author Contributions: OK

**Funding: OK

**Data Availability Statement: OK

**Conflicts of Interest: OK

**References:

-Please capitalize all the nouns of the title or only the first noun. For example, the authors should write “Biochar and effective microorganisms promote Sesbania cannabina growth and soil quality in the coastal saline-alkali soil of the Yellow River Delta, China” or “Gene Expression Profiling of Plants under Salt Stress.” They cannot keep both systems at the same time in the References.

Reviewer 2 Report

Comments and Suggestions for Authors

Dear Authors

Plese see my comments in the added file.

Comments on the Quality of English Language

I have no specific comments 

Reviewer 3 Report

Comments and Suggestions for Authors

Research is relevant and interesting.

1. Methods of statistical analysis need to be described more detailed.

2. The conclusions could be more concrete with a proposal for the future.

3. I recommend not to use old literature sources.

Round 2

Reviewer 2 Report

Comments and Suggestions for Authors

Please see added file of comments

Author Response

Thank you very much for taking the time to review this manuscript. In our Round 1 response, we provided detailed explanations of the questions in the attachment, and we apologize for not clearly indicating these in the revised manuscript.

Please see the attachment for the Round 2 point-by-point response to the reviewer’s comments.

Round 3

Reviewer 2 Report

Comments and Suggestions for Authors

I accept the Third version after your additions and explanations.